# Life Table Construction under Different Temperatures and Insecticide Susceptibility Analysis of *Uroleucon formosanum* (Hemiptera: Aphididae)

**DOI:** 10.3390/insects13080693

**Published:** 2022-08-01

**Authors:** Tian-Xing Jing, Chu-Chu Qi, Ao Jiao, Xiao-Qiang Liu, Shuai Zhang, Hong-Hua Su, Yi-Zhong Yang

**Affiliations:** 1College of Horticulture and Plant Protection, Yangzhou University, Yangzhou 225009, China; jtx1298073671@163.com (T.-X.J.); qcc821534982@163.com (C.-C.Q.); ja19131039056@163.com (A.J.); shuaizhang@yzu.edu.cn (S.Z.); susugj@126.com (H.-H.S.); 2Environment and Plant Protection Institute, Chinese Academy of Tropical Agricultural Sciences, Haikou 571101, China; xqlcoin@163.com

**Keywords:** *Uroleucon formosanum*, life table, insecticide bioassay, pest control

## Abstract

**Simple Summary:**

Aphids are major crop pests worldwide, and in China, *Uroleucon formosanum* is a common aphid pest of lettuce. However, there is little basic and applied information on the control of this pest. To obtain the basic information of this pest, a life table of *U. formosanum* under different temperatures was constructed. Also, the susceptibility of *U. formosanum* to six common-used insecticides (chlorpyrifos, abamectin, beta-cypermethrin, imidacloprid, nitenpyram, and thiamethoxam) was evaluated. Results showed that *U. formosanum* was not suitable to a lower temperature (17 °C), and *U. formosanum* was relatively sensitive to all six test insecticides. These data may help us to develop integrated management strategies for better population control of *U. formosanum*.

**Abstract:**

*Uroleucon formosanum* is an important aphid pest of lettuce, but basic information on its biology is scarce. In this study, effects of three constant temperatures (17, 21, and 25 °C, simulating the mean temperature range in greenhouses) on the development and fecundity of *U. formosanum* were analyzed by constructing a life table. *U. formosanum* could develop and reproduce under all three temperatures, but the survival rate, development, and fecundity of *U. formosanum* were affected by temperature. The intrinsic rate of increase was lowest at 17 °C (0.17) and it was significantly less than at 21 °C (0.20) and 25 °C (0.23). Furthermore, *U. formosanum* had the lowest finite rate of increase (1.19) and the largest mean generation time (20.21) at 17 °C. These results mean that *U. formosanum* is less adapted to the lower temperatures (17 °C) among these three set temperatures. To screen insecticides for control, susceptibility of *U. formosanum* to six insecticides including chlorpyrifos, abamectin, beta-cypermethrin, imidacloprid, nitenpyram, and thiamethoxam was evaluated. *U. formosanum* was relatively sensitive to all six test insecticides. Chlorpyrifos had the highest toxicity to *U. formosanum* (LC_50_ = 3.08 mg/L). These data may help to develop integrated management strategies for better population control of *U. formosanum.*

## 1. Introduction

Lettuce (*Lactuca sativa* L.) is a leafy vegetable that is planted worldwide. In 2020, world production of lettuce exceeded 27 million tons, and China produced the highest amount at 14.3 million tons (Food and Agriculture Organization of the United Nations, FAO). Major economic losses to lettuce are caused by pests such as beet armyworm *Spodoptera exigua*, western flower thrips *Frankliniella occidentalis*, cabbage looper *Trichoplusia ni*, and several aphid species [1,2,3,4].

Aphids include many important pests of agriculture, horticulture, and forestry. They can directly damage plants by sap-feeding, and indirectly damage plants by honeydew secretion or virus transmission [2,5]. On lettuce, *Nasonovia ribisnigri* (Mosley), *Macrosiphum euphorbiae* (Thomas), and *Myzus persicae* (Sulzer) are frequent aphid pests [6]. Among these aphids, *N. ribisnigri* is considered to be the main pest on lettuce and is often referred to as the lettuce aphid. There are many studies on integrated control of this insect, combining chemical insecticide applications and the use of biological control [3,6,7]. However, *N. ribisnigri* is mainly distributed in the temperate regions of Europe, North and South America, and Australia [4,6,7]. In China. studies on *N. ribisnigri* are limited because *N. ribisnigri* was previously not found there and was listed as a quarantine pest [4]. In Eastern Asia, another pest aphid, *Uroleucon formosanum*, damages lettuce plants. The *Uroleucon* genus contains many species that have been recorded to feed on the Asteraceae without host alternation [8,9]. In Eastern Asia, *U. formosanum* is commonly recorded feeding on Asteraceae species in the genera *Lactuca, Ixeris, Picris*, and *Sonchus* [10]. *U. formosanum* extracts leaf sap by suction and reduces the yield and quality of the vegetables it attacks. In addition to causing direct damage, lettuce aphids also cause commercial damage due to the presence of living aphids, which contaminate the lettuce and render it unacceptable to consumers [6]. Occurrence of *U. formosanum* was very serious in greenhouses in Yangzhou, and there is an urgent need to control this pest. However, there is little information available on its basic biology. Most studies on *U. formosanum* have focused on its taxonomy, morphology, host range, phylogeny, and evolution [9,10,11]. The lack of basic biological information has made it more difficult to develop effective control strategies.

The life table is a powerful tool used to collect information on the growth, survival, reproduction, and intrinsic rate of increase of insect populations. This basic information can help us to establish more effective management tactics [12,13]. In this study, the life table parameters of *U. formosanum* were studied under constant temperatures (lettuce greenhouse temperatures). Additionally, six commonly used insecticides were used in bioassays to determine the most effective insecticides for *U. formosanum* control. This study provides basic information on *U. formosanum* and may aid in the development of integrated management strategies for aphid control. 

## 2. Materials and Methods

### 2.1. Insect Colony

*U. formosanum* aphids were originally collected from an *L. sativa* field in Yangzhou, China in 2020 and then maintained on fresh leaves of *L. sativa* in climate incubators under the conditions of 21 ± 1 °C, 70–75% relative humidity and a 14:10 h (L:D) photoperiod.

### 2.2. Life Table Construction under Constant Temperature

Based on the common temperature range in lettuce greenhouses (17–25 °C), three constant temperatures (17, 21 and 25 °C) were established to study the nymph development and adult reproduction of *U. formosanum*. For observation of nymph and adult development, five *U. formosanum* adults were transferred onto a lettuce leaf disk and then placed upside on a wet filter paper in a plastic dish (35 mm diam.). Six hours later, one newborn nymph was retained, and the other aphids (both adults and nymphs) were removed using a soft brush. The molting of newborn *U. formosanum* was observed and recorded every 24 h until all aphids were dead. Forty newborn nymphs were used to conduct the life table analysis. During the reproductive period, the newborn nymphs were counted and then removed using a soft brush every 24 h until the adult died.

### 2.3. Insecticide Bioassay

To evaluate the efficiency of commonly used insecticides labeled for aphid control, six insecticides, chlorpyrifos (97% purity, Qixing Pesticide Company, Yuncheng, China), abamectin (95% purity, Qixing Pesticide Company, Yuncheng, China), beta-cypermethrin (96% purity, Qixing Pesticide Company, Yuncheng, China), imidacloprid (96% purity, Suling Pesticide & Chemical Company, Yangzhou, China), nitenpyram (95% purity, Liben Pesticide & Chemical Company, Lianyungang, China), and thiamethoxam (95% purity, Qixing Pesticide Company, Yuncheng, China), were chosen to conduct bioassays using the leaf-dipping method [14]. Pure insecticides were first diluted with acetone to 10,000 mg/L stock solutions. For bioassays, stock solutions were diluted to serial concentrations (based on the results of preliminary testing) using ultrapure water containing 0.05% Triton X-100 (Sangon Biotech, Shanghai, China). Lettuce leaf disks were immersed in insecticide solution for 10 s. After natural drying (approximately 10 min), 20 adult aphids (three days old) were transferred onto each treated leaf disk. After 24 h, the aphids that were immobile or unable to walk after being gently prodded with a soft brush were considered to be dead and were recorded. Aphids treated with 0.05% Triton X-100 water were used as the control to assess natural mortality. The experiments were replicated three times for each insecticide and its dilutions.

### 2.4. Statistical Analysis

Raw life history data were analyzed based on the theory of an age-stage, two-sex life table. The life history data, survival, longevity and fecundity of *U. formosanum* under different constant temperatures were analyzed using the TWOSEX-MS Chart program [15]. The age-stage-specific survival rate, age-specific survival rate (*l_x_*), age-specific fecundity (*m_x_*), age-specific maternity (*l_x_m_x_*), age-stage-specific life expectancy (*e_xj_*) were calculated [15]. The population parameters, intrinsic rate of increase (*r*), net reproductive rate (*R*_0_), finite rate of increase (*λ*) and the mean generation time (*T*) of *U. formosanum* under different constant temperatures were analyzed. These parameters were calculated using the following formulas. The means and standard errors of developmental duration, as well as the fecundity and population parameters, were analyzed using bootstrapping methods in TWOSEX-MS Chart program with 10,000 resamples [16]. For the insecticide bioassay, adjusted mortality data were subjected to probit analysis and the median lethal concentration (LC_50_), 95% confidence interval, and slope were calculated using SPSS (v. 20).

## 3. Results

The durations of all developmental stages (four nymph stages and the adult stage) of *U. formosanum* under different temperatures are listed in Table 1. The shortest total longevity (17.18 ± 1.33 d) duration was observed at 25 °C with a significant difference compared to 17 °C (19.05 ± 1.69 d) and 21 °C (19.08 ± 1.11 d). At 25 °C, *U. formosanum* had reduced total longevity but the longest adult longevity (13.24 ± 1.09 d) and the corresponding shortest preadult duration (6.25 ± 0.26 d). The age-stage-specific survival rate is shown in Figure 1A. The survival of *U. formosanum* nymphs reared at 21 °C (92.5%) was higher than at 17 °C (80.0%) and 25 °C (85.0%). The age-stage life expectancy (*e_xj_*) indicated the time that an aphid on age *x* and stage *y* is expected to live. The *e_xj_* curve showed that *U. formosanum* has the largest and shortest life expectancy at 17 °C and 25 °C, respectively (Figure 2).

### 3.1. Effect of Temperature on the Fecundity of Uroleucon formosanum

APRP (adult prereproductive period) and TPRP (total prereproductive period) decreased with increasing temperature and the shortest APRP (1.38 ± 0.09 d) and TPRP (9.19 ± 0.18 d) were found at 25 °C. These values were significantly different compared to the longest APRP (2.07 ± 0.17 d) and TPRP (13.83 ± 0.18 d) at 17 °C. The greatest fecundity (32.57 ± 1.84 nymphs per female) of *U. formosanum* occurred at 21 °C, and the lowest fecundity (25.31 ± 1.83 nymphs per female) occurred at 25 °C. The survival rate (*l_x_*) and age-specific fecundity (*m_x_*) of *U. formosanum* under the three temperatures are shown in Figure 1B. The *m_x_* curve was relatively flat at 21 °C compared to 17 °C and 25 °C, and reached a peak at the age of 22 d (4.92 offspring/d). The *m_x_* curve of 17 °C and 25 °C had larger fluctuations, and the highest age-specific fecundity occurred at the age of 20 d (5.67 offspring/d) and 12 d (5.89 offspring/d), respectively. In combination with the survival rate *l_x_*, the maximum *l_x_m_x_* values were at the age of 20 d (2.44 offspring), 12 d (3.19 offspring), and 12 d (4.30 offspring) at 17, 21, and 25 °C, respectively.

### 3.2. Life Table Parameters

The life table parameters of *U. formosanum,* reared at different temperatures, are presented in Table 2. The intrinsic rate of increase (*r*) of *U. formosanum* increased with temperature, and the highest *r* (0.23 ± 0.00) occurred at 25 °C. The finite rate of increase (*λ*) showed the same tendency as *r* and the highest *λ* (1.26 ± 0.00) of *U. formosanum* occurred at 25 °C. *U. formosanum* reared at 17 °C had the lowest *r* (0.17 ± 0.01), *λ* (1.19 ± 0.01) and the largest mean generation time (*T*) (20.21 ± 0.49). The mean generation time (*T*) decreased with temperature, and the shortest *T* (13.81 ± 0.47) occurred at 25 °C. The largest net reproductive rate *R*_0_ (32.58 ± 1.84) occurred at 21 °C, while *U. formosanum* had the lowest *R*_0_ at 25 °C.

### 3.3. Insecticide Bioassay

Chlorpyrifos, abamectin, beta-cypermethrin, imidacloprid, nitenpyram and thiamethoxam were used to conduct bioassays of *U. formosanum*. The results, including the determination of LC_50_ values, are shown in Table 3. *U. formosanum* had different susceptibility to these six insecticides. The order of toxicity of the insecticides to *U. formosanum*, from high to low, was as follows: chlorpyrifos, abamectin, beta cypermethrin, imidacloprid, nitenpyram, and thiamethoxam; the LC50 values of these insecticides were 3.08, 14.33, 17.67, 30.67, 36.57, and 46.89 mg/L, respectively. Chlorpyrifos had the highest toxicity to *U. formosanum*.

## 4. Discussion

Aphids are major crop pests worldwide, and in China, *U. formosanum* is a common aphid pest of lettuce. However, there is little basic and applied information on the control of this pest. Because China produces considerable amounts of lettuce, it is important to understand the features of this main aphid pest of lettuce [17]. To improve production efficiency, most lettuce is planted in greenhouses with temperatures which range from 17–25 °C (mean = 21 °C) [18]. These temperatures are suitable for lettuce growth, but also promote the development of pests on lettuce. Temperature is a crucial factor affecting insect development, fecundity behavior and fitness [19,20]. Insect life tables have been extensively used to analyze and understand the effects of environmental factors, including temperature, on the survival, growth, and reproduction of insects [21,22,23]. Life tables have also been widely applied in the study of insect population ecology and pest management. However, life table research on *Uroleucon* is limited. Temperature effects on *Uroleucon ambrosiae* were reported, and weather parameter effects on the population dynamics of *U. compositae* were described [18,24]. Our study demonstrated that *U. formosanum* can develop and reproduce at constant temperatures ranging from 17–25°C, and that the temperature has a significant effect on the development, reproduction, and population parameters of *U. formosanum*.

The total longevity of *U. formosanum* was longer at the lower temperature (17 °C), and the nymph longevity (from first instar to fourth instar) was also longer at 17 °C. This phenomenon was also found in most aphid species, including another lettuce aphid *U.*
*ambrosiae* (Table 2). However, the longest adult longevity occurred at 25 °C. APRP and TPRP were temperature-dependent from 17–25 °C. They decreased with temperature, and this tendency has been reported in many other insects including several aphid species [13,18,22]. The role of temperature in accelerating the population growth of herbivores by shortening development time has been described in numerous species [13]. Temperature has significant effects on development rates and behavior of insects. It is a systematic work to study the effect of temperature (both constant and fluctuating temperature) on this pest. In this study, three constant temperatures were set to obtain some basic biological information of *U. formosanum.* Fluctuating temperature is more realistic than constant temperature in natural conditions, and models built under fluctuating temperature may improve the predictions of the establishment and spread of insects [25,26,27]. This will be our further research: to study the development of *U. formosanum* under a wider set of fluctuating temperatures.

Although *U. formosanum* had the longest adult stage at 25 °C, it had the lowest fecundity (25.31 ± 1.83 nymphs per female) among the three temperatures. The highest fecundity (32.58 ± 1.84 nymphs per female) occurred at 21 °C. While at 17 °C, *U. formosanum* had the lowest intrinsic rate of increase, a finite rate of increase, and the largest mean generation time. Based on these parameters, *U. formosanum* was not suitable to 17 °C among these three constant temperatures.

The most important life table parameters for describing population growth were analyzed in this study. The intrinsic rate of increase *r* and finite rate of increase *λ* increased with temperature. Increases of *r* and *λ* have been recorded in aphids at a range of temperatures (from 20–25 °C) including *U. ambrosiae* and *Aulacorthum solani* [18,28]. In contrast to the two parameters above, the net reproductive rate *R*_0_ was not temperature-dependent and reached a peak at 21 °C. From these results, 21 °C appears to be the most optimal temperature for *U. formosanum*. In addition, at 21 °C, the age-specific fecundity (*m_x_* curve) stability was better than at the other two temperatures.

There is little information, especially related to control strategies, about *U. formosanum* on lettuce. Using natural enemies to control aphids on lettuce is a feasible method. Conti et al. compared the development of the parasitoid *Praon volucre* feeding on *Macrosiphum euphorbiae* (the common host of *P. volucre*) and *U. ambrosiae* under a constant temperature 22 ± 1 °C. It seems that *P. volucre* was a good candidate for biocontrol of aphids on lettuce [29]. Similarly, Shrestha et al. reported that *Aphelinus abdominalis* has the potential to be used against *N. ribisnigri* on lettuce [30]. However, it was not clear if these natural enemies can be used to control *U. formosanum.* Therefore, several insecticides commonly used to control aphids were selected to conduct bioassays to screen for efficient insecticides. *U. formosanum* was most sensitive to chlorpyrifos, with an LC_50_ of 3.08 mg/L. For many aphids, including *Rhopalosiphum padi, Sitobion avenae*, and *Metopolophium dirhodum*, the LC_50_ of chlorpyrifos is less than 10 mg/L [31,32]. These data indicated that *U. formosanum* has minimal resistance to the organophosphate chlorpyrifos. Although organophosphates were frequently used in the past and selected for resistance in target pests, it has been reported that resistance to organophosphates can decline rapidly over several years if the use of these insecticides is discontinued [33]. Abamectin is a microbial-derived pesticide frequently used to control insects, mites and nematodes owing to its high efficiency, low residues, broad-spectrum activity, and relative safety to humans, animals, and the environment [34,35]. For many of the aphids studied, most LC_50_ values of abamectin were at a low level (less than 15 mg/L), and the LC_50_ of *M. dirhodum* was 1.60 mg/L (LC_50_) [32]. In *U. formosanum*, the LC_50_ of abamectin was also less than 15 mg/L (14.33 mg/L), which indicates that abamectin can also be an efficient insecticide to control this aphid species. The pyrethroid beta-cypermethrin has been applied to control agricultural pests for over 30 years because of its high efficacy, safety, broad spectrum of activity and low human toxicity [36]. However, resistance to pyrethroids has been reported in many insect pests. For *M. dirhodum* and *R. padi*, the susceptibility baseline of beta-cypermethrin was 0.52 mg/L and 1.2 mg/L, and LC_50_ values of most field populations ranged from 2.6–22.8 mg/L [31,32]. In contrast to the bioassay results of *M. dirhodum* and *R. padi* field populations, *U. formosanum* had normal susceptibility to beta-cypermethrin. Furthermore, beta-cypermethrin continues to be an efficient insecticide for control of *U. formosanum*. Neonicotinoids are commonly used against sap-feeding insect pests [32]. We chose three neonicotinoids (imidacloprid, thiamethoxam and nitenpyram) for bioassay evaluation. The LC_50_ values of these three insecticides, from low–high in order, were 30.67 mg/L (imidacloprid), 36.57 mg/L (nitenpyram) and 46.89 mg/L (thiamethoxam). Neonicotinoids are the main insecticides used to control aphids. With the extensive application of these insecticides, high resistance has been reported in the field populations of several aphid species [5]. Compared to the susceptibility baseline of these insecticides to other aphids, *U. formosanum* may have developed low resistance (approximately 20–30 fold) to neonicotinoids. Though the resistance level is not high, and neonicotinoids can still be used to control *U. formosanum*, there is a risk that *U. formosanum* may ultimately develop high resistance to neonicotinoids. Based on the bioassay results, all these insecticides can still be used to efficiently control this aphid pest. However, resistance remains a problem for sustainability of pest control in instances where chemical insecticides continue to be frequently used. Rotation or combination use of different kinds of insecticides may help slow the development of resistance. The use of neonicotinoids should be prudent.

## 5. Conclusions

The development and fecundity of *U*. *formosanum* on lettuce were investigated under three constant temperatures (17, 21 and 25 °C). Additionally, the susceptibility of *U. formosanum* to six commonly used insecticides (chlorpyrifos, abamectin, beta-cypermethrin, imidacloprid, nitenpyram, and thiamethoxam) was evaluated. Results indicated that *U. formosanum* was not suitable to lower temperature (17 °C) among these set temperatures, and that *U. formosanum* was relatively sensitive to all six test insecticides. These data may help us to develop integrated management strategies for better population control of *U. formosanum.*

## Figures and Tables

**Figure 1 insects-13-00693-f001:**
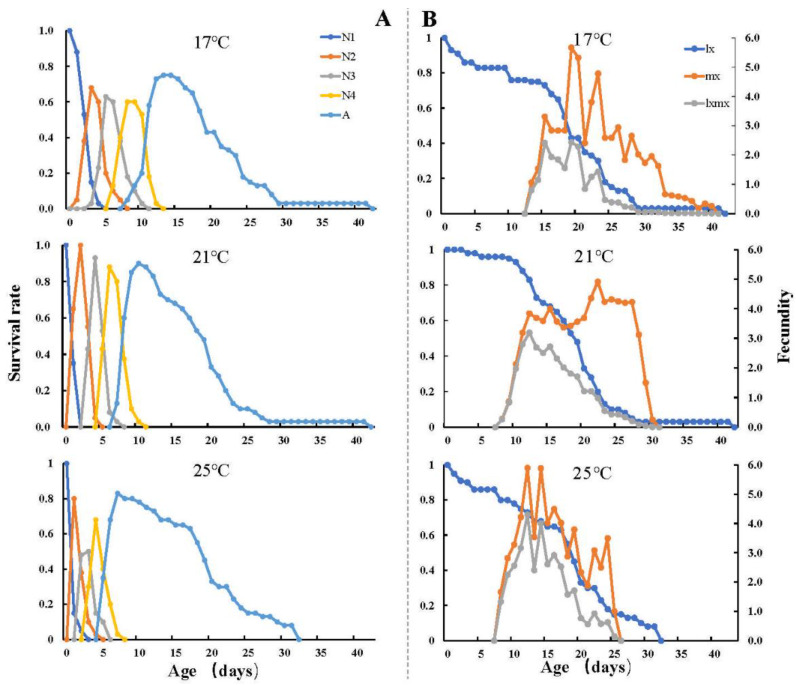
The age-stage-specific survival rate and age-specific fecundity of *Uroleucon formosanum* under different temperatures. (**A**) the age-stage-specific survival rate of *U. formosanum* under different temperatures. (**B**) the age-specific fecundity of *U. formosanum* under different temperatures. N1, first instar nymph; N2, second instar nymph; N3, third instar nymph; N4, fourth instar nymph; A, adult; lx, age-specific survival; mx, age-specific fecundity; lxmx, age-specific maternity.

**Figure 2 insects-13-00693-f002:**
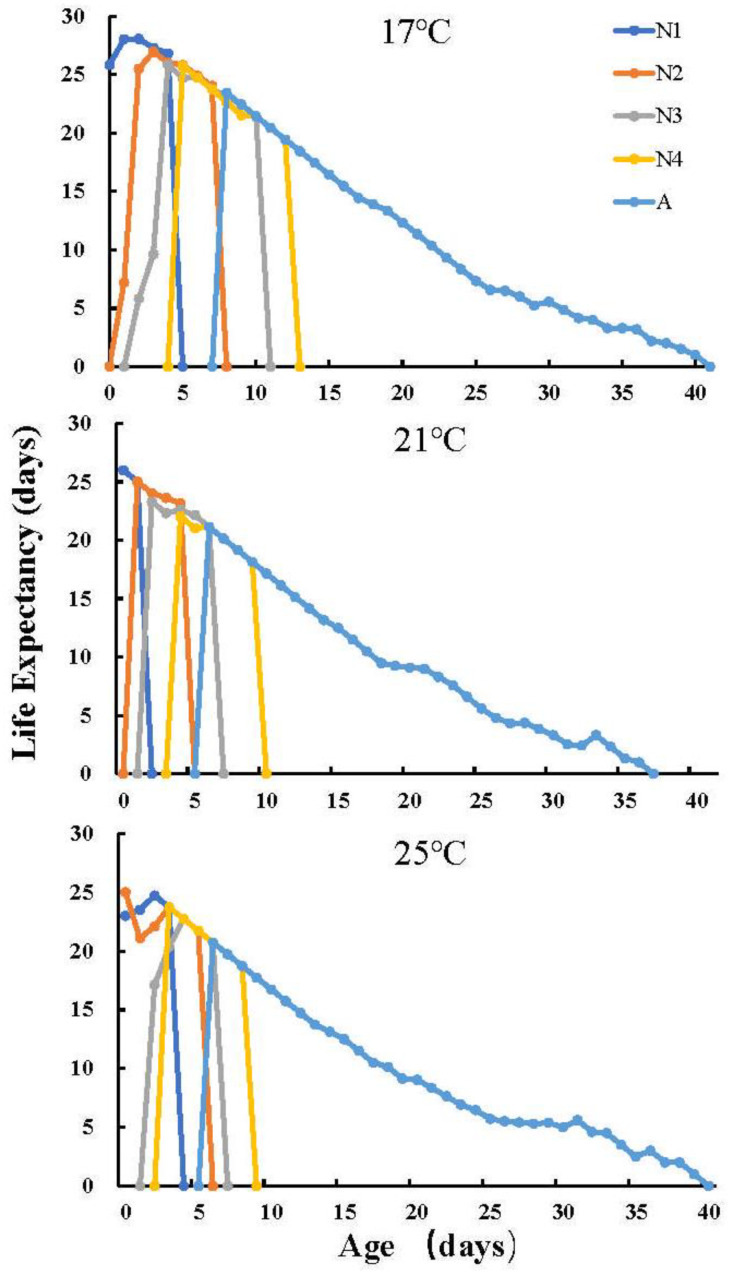
Age-stage-specific life expectancy (*e_xj_*) of *Uroleucon formosanum*. N1, first instar nymph; N2, second instar nymph; N3, third instar nymph; N4, fourth instar nymph; A, adult.

**Table 1 insects-13-00693-t001:** Developmental duration and fecundity of *Uroleucon formosanum* under different temperatures.

Stage	*Uroleucon formosanum*		
	17 °C	21 °C	25 °C
First instar	2.65 ± 0.18 a	1.35 ± 0.08 b	1.78 ± 0.12 b
Second instar	2.16 ± 0.12 a	2.25 ± 0.09 a	1.64 ± 0.08 b
Third instar	2.67 ± 0.13 a	2.00 ± 0.09 b	1.49 ± 0.13 c
Fourth instar	2.88 ± 0.22 a	2.74 ± 0.10 a	1.85 ± 0.09 b
Preadult duration	9.33 ± 0.50 a	8.15 ± 0.20 b	6.25 ± 0.26 c
Adult longevity	12.87 ± 1.49 ab	11.68 ± 1.00 b	13.24 ± 1.09 a
Total longevity	19.05 ± 1.69 a	19.08 ± 1.11 a	17.18 ± 1.33 b
APRP	2.07 ± 0.17 a	1.83 ± 0.09 a	1.38 ± 0.09 b
TPRP	13.83 ± 0.18 a	11.28 ± 0.15 b	9.19 ± 0.18 c
Fecundity (offspring per female)	31.68 ± 3.07 a	32.58 ± 1.84 a	25.31 ± 1.83 b

For *Uroleucon formosanum,* means and standard error (SE) were estimated by the bootstrap technique with 10,000 resampling using TWOSEX-MSChart. Means in the same row followed by a different letter indicates significant differences at *p* < 0.05. APRP, adult prereproductive period; TPRP, total prereproductive period.

**Table 2 insects-13-00693-t002:** Life table parameters of *Uroleucon formosanum* under different temperatures.

Temperature (°C)	*T*	*R* _0_	*r*	*λ*
17	20.21 ± 0.49 c	31.68 ± 3.07 b	0.17 ± 0.01 a	1.19 ± 0.01 a
21	17.35 ± 0.06 b	32.58 ± 1.84 b	0.20 ± 0.01 b	1.22 ± 0.00 b
25	13.81 ± 0.47 a	25.31 ± 1.83 a	0.23 ± 0.01 c	1.26 ± 0.00 c

The means, standard errors (SE) and significant differences were analyzed using bootstrapping methods in TWOSEX-MSChart. 100,000 replications were used in the bootstrapping procedures. Means in the same row followed by a different letter indicate significant differences at *p* < 0.05. *T*, the mean generation time; *R*_0_, net reproductive rate; *r*, intrinsic rate of increase; *λ*, finite rate of increase.

**Table 3 insects-13-00693-t003:** Susceptibility of *Uroleucon formosanum* to six insecticides.

Insecticide	Slope ± SE	LC_50_ (mg/L)	95% CI	n
Chlorpyrifos	0.55 ± 0.11	3.08	2.35–3.60	300
Abamectin	0.09 ± 0.02	14.33	10.74–17.63	300
Beta-cypermethrin	0.03 ± 0.01	17.67	9.17–28.97	300
Imidacloprid	0.02 ± 0.01	30.67	12.58–46.71	300
Nitenpyram	0.02 ± 0.01	36.57	28.20–43.44	300
Thiamethoxam	0.02 ± 0.00	46.89	31.11–65.07	300

SE, standard error; LC_50_, lethal concentration of 50%; 95% CI, 95% confidence interval; n, number of aphids used in bioassay.

## Data Availability

Data presented in this study are available in the article.

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
