# Peer review of "Life Table Construction under Different Temperatures and Insecticide Susceptibility Analysis of Uroleucon formosanum (Hemiptera: Aphididae)"

_insects, 2022, doi:10.3390/insects13080693_

Round 1

Reviewer 1 Report

The study ''Life table construction under different temperatures and insecticide susceptibility analysis of Uroleucon formosanum (Hemiptera: Aphididae)'' is interesting, well designed and could be considered with minor corrections. Overall the study is well written and easy to understand for readers.

Line 17, 'and' should not be italic.

Line 273, Ref 1 need to includes the authors name correctly. Check all references carefully.

Carefully read the whole manuscript to avoid any spell mistakes.

Author Response

Comment: The study ''Life table construction under different temperatures and insecticide susceptibility analysis of Uroleucon formosanum (Hemiptera: Aphididae)'' is interesting, well designed and could be considered with minor corrections. Overall, the study is well written and easy to understand for readers.

Response: We really appreciate your valuable suggestions which will improve the quality of this paper to a large extent.

Comment: Line 17, 'and' should not be italic.

Response:The italic “and” was modified as “and”.

Comment: Line 273, Ref 1 need to includes the authors name correctly. Check all references carefully.

Response: Ref 1 was modified as following, and we check all references.

“Haile, F.J.; Kerns, D.L.; Richardson, J.M. and Higley, L.G. Impact of insecticides and surfactant on lettuce physiology and yield. J Econ Entomol 2000, 93, 788-794.”

Comment: Carefully read the whole manuscript to avoid any spell mistakes.

Response: we checked the whole manuscript to avoid spell mistakes.

Reviewer 2 Report

The authors have graphed and presented their results clearly, drawing some attention to the implications of their findings. I found the study of interest and a good contribution to the knowledge of bioecology of aphid pests. The methods used are appropriate for the objectives of the work and, in general, well depicted. The resulting figures are sufficient, informative, and of good quality helping to follow the reasoning throughout the manuscript.

Nevertheless, the authors need to decide if the focus is on aphid biology or insecticide screening efficacy against these pests. The article would be much more compelling if the focus is on biology of this pest. The insecticide part is out of the scope of the paper and could be omitted.

Also, the Intro and Discussion provide no insight on how this MS relates to the various other ones cited in the text or concerns that have been raised by other researchers. This article should provide details on all these fronts to provide the proper context for the work. Authors do not present any hypotheses or expectations that could be connected to previous studies; adding these details will improve the paper. The authors should clearly explain WHY THE STUDY WAS DONE, WHY IT WAS IMPORTANT, and HOW IT FITS WITH OTHER STUDIES. It should be clear and concise. The intro should also include what outcome(s) they expect, and how it would help support or refute their hypotheses or answer their questions.

My final concern is that the authors are extrapolating the applicability of their results beyond what the design supports. Extrapolation of life table data was based on three sets of highly artificial constant temperatures. Some of the authors statement would be much stronger if they tie and compare their work to the body of literature that has built up from rearing pests and their natural enemies at a wider set of realistic fluctuating temperatures. This is not to diminish the data gathered in this study, they are of value. But it is important for the authors not to overgeneralize, and to warn the reader against doing so as well. Some suggestions might include but are not limited to J. Econ. Entomol. 113: 633-645., J. Econ. Entomol. 112: 1560-1574., J. Econ. Entomol. 112:1062-1072., etc. etc. Adding these details will improve the paper in my opinion.

Good luck!

Round 2

Reviewer 2 Report

The authors have done a fine job addressing all of my original concerns. I have no additional suggestions. Thank you.